# Acceptability of a Mobile Application in Children’s Oral Health Promotion—A Pilot Study

**DOI:** 10.3390/ijerph18063256

**Published:** 2021-03-22

**Authors:** Kirsi Rasmus, Antti Toratti, Saujanya Karki, Paula Pesonen, Marja-Liisa Laitala, Vuokko Anttonen

**Affiliations:** 1Research Unit of Oral Health Sciences, University of Oulu, P.O. Box 5281, 90014 Oulu, Finland; Kirsi.rasmus@student.oulu.fi (K.R.); antti.toratti@oulu.fi (A.T.); marja-liisa.laitala@oulu.fi (M.-L.L.); vuokko.anttonen@oulu.fi (V.A.); 2Infrastructure for Population Studies, Faculty of Medicine, University of Oulu, P.O. Box 5281, 90014 Oulu, Finland; paula.pesonen@oulu.fi; 3Medical Research Center, Oulu University Hospital, University of Oulu, P.O. Box 5281, 90014 Oulu, Finland

**Keywords:** oral health promotion, mobile applications, children

## Abstract

The aim of this pilot study was to investigate the acceptability of an oral health-related mobile application developed for young children based on the feedback given by the children and their parents. Another aim was to evaluate the self-reported change in children’s oral health behaviors during a short test period. The application—a virtual pet integrated into a child’s daily routines—aimed to promote oral hygiene and dietary behaviors in children. A total of 36 4–12-year-old voluntary children were given a mobile phone with the installed application. After the 5-week testing period, the feasibility of the application and possible changes in the children’s oral health behaviors were asked using an electronic questionnaire. Most of the children considered the application clear (n = 34), amusing (n = 31), and useful (n = 29). The children’s tooth brushing manners improved both qualitatively and quantitatively: the time used for tooth brushing increased and the children learned how to brush different tooth surfaces. Mobile applications can be fun and useful in oral health promotion; while playing, children can learn good oral health-related behaviors. Mobile applications integrate oral health promotion into children’s daily environment and routines.

## 1. Introduction

The oral health of children is part of general health and a key to health in adulthood. Indeed, oral infections have been shown to be associated with systemic disease in adulthood [1]. Untreated dental caries, one of the oral diseases, still remain a major public health challenge affecting 573 million children globally [2]. The need for restorative treatment is also high among children with special health care needs [3]. Early childhood caries (ECC), a severe form of dental caries in children, affects a remarkable proportion of children in low-, middle-, and high-income countries [4]. In 2006, Sheiham proposed a clear association between untreated dental caries, quality of life, and poor growth in young children [5].

In 2020, approximately 3.5 billion people owned a smart phone, and the number of mobile phone users was expected to be 3.8 billion in 2021 [6]. The growth in the number of smart phones is the fastest in developing countries such as China, India, and Latin America. Information technology, mobile applications, and games can be used as a means of health promotion, and several patient-focused mobile applications are already available [7].

The use of reminders via mobile text messages (SMS, short message service) were found to be effective in promoting and motivating 18- to 24-year-old adults, and regular tooth brushing increased from 51% to 73% during the test period of 12 weeks [8]. Similarly, another mobile phone application was found to encourage its users to brush for two minutes and also allowed users to set reminders for brushing or changing their toothbrush [9]. It also helped to educate the users on the prevention of dental caries and periodontal diseases by linking it to animated videos published on YouTube [9]. Aljafari and colleagues also found that oral health promotion using a computer game could be as satisfactory and effective as one-to-one education in improving knowledge in children at high risk for dental caries [10]. Systematic reviews have shown that oral health promotion programs are able to improve clinical outcomes among children and adolescents [11,12]. 

Children and adolescents are used to living in contexts where smart phones and tablet computers have always been available. It was reported that children as young as six understand digital technology better than adults [13]. The use of a digital platform for improving oral health in the long-term can be one behavioral theory-based strategy. Therefore, smart phone applications have the potential to deliver new information to children, for example, good oral health behaviors.

Behavioral change is an important aspect of oral health promotion [14]. Self-efficacy is considered to be an individual’s belief in their capacity to execute behaviors necessary to provide specific performance for attainment. In this model, mastery experiences, vicarious experiences, verbal persuasion, and emotional arousal can be the sources of information to be motivated for goal attainment [15]. There is evidence concerning the influence of theory-based behavioral change on dental caries reduction [16]. The parental role is crucial in adopting healthy habits including oral hygiene. A systematic review reported that maternal self-efficacy was associated with dental caries in children [17]. 

The aim of this pilot study was to investigate the acceptability of an oral health-related mobile application developed for young children based on the feedback given by the children and their parents. Another aim was to evaluate the self-reported change in children’s oral health behaviors during a short test period of five-weeks. The null hypothesis for this study is that a mobile phone application designed for preschool as well as primary school children is not acceptable for use and it cannot improve oral health behaviors in a short test period.

## 2. Materials and Methods

### 2.1. Study Design

This pilot study was conducted among children from kindergartens and primary schools and their caregivers in the city of Oulu, Finland, from September–November 2016 over a period of five weeks. 

### 2.2. Study Participants

This study was comprised of 36 voluntary children from kindergartens and primary schools and their caregivers. The participants were recruited via local school/day care center principals. Therefore, the participation rate was not calculated. The target group was intentionally small because the aim was to further develop the mobile application after the pilot test period based on the feedback and comments received from the study participants.

### 2.3. Applications and Devices

The Denny^®^ (Ikoni Innovations Company Oy, Oulu, Finland) mobile application, comprised of “Denny the Tooth^®^ and Denny Timer^®^” (Figure 1a,b), were developed in collaboration with the university and two companies in 2016 aiming to find digital solutions for oral health promotion. The expertise on dental health and oral health promotion came from experts in the field.

Denny the Tooth^®^ is a virtual pet—a tooth that lives in children’s mobile devices. It follows the same rhythm as the child: it wakes up in the morning, as does their owner, it consumes whatever its owner wants to feed it, and its teeth can be brushed, for example, in the morning and in the evening before bedtime while the child brushes his or her own teeth.

Denny Timer^®^ is also a tooth brushing assistant: it gives children advice on how and how long to brush their teeth. The Denny^®^ applications are designed for children aged five to 12 years. The applications were developed for both Android and iOS operating systems, and at the time of the study, they were available on Google Play Store^®^ (Alphabet Inc., Mountain View, CA, USA) and AppStore^®^ (Apple Inc., Sunnyvale, CA, USA) free of charge.

The smart phones used in this study were provided to all participants. The phone model was Alcatel Pixi 4^®^ (TCL Technology, Shenzhen, China/Nokia Oyj, Espoo, Finland) with both Denny^®^ applications installed. The phones did not include SIM cards and could not be used for making phone calls.

During the development process, the application was tested among some children from the target age groups. At baseline, the participating children and their parents took part in an introductory and oral health-related information session lasting about 10 min.

After the lecture, the children were given the phones with pre-installed applications and then given some time to become acquainted with the Denny^®^ mobile phone application (both Denny the Tooth^®^ and Denny Timer^®^) without any instructions. 

Two-weeks after the recruitment of the study participants, they were sent a reminder about the Denny^®^ mobile phone application. After five weeks, all phones were returned, and while doing so, the participants (i.e., caregiver) received a link to a Webropol-based survey.

### 2.4. Study Outcomes

The survey consisted of a questionnaire to determine the child’s mobile phone ownership, the responsible person for selecting mobile phone applications, pricing of the mobile applications (free/paid), and willingness to pay a small non-recurring fee for a useful application that offered child education/entertainment. Caregivers who responded ‘no’ to the latter part were further asked to comment freely. In addition, the questionnaire also included the caregivers’ perceptions of the Denny^®^ mobile phone application (*useful/clear/enough content/useless/fun/boring*) and whether any changes in oral health behaviors could be seen while using the application (*Did the application motivate the child in tooth brushing/avoid snacking, did the behaviors change and if they did, in which way*). The options for all questions were *Yes/No/I can’t say*. There was also questions on how the Denny applications influenced the child’s oral health routines (*increased the frequency of tooth brushing in the morning/evening*). After the questions, the caregivers were also requested to give suggestions on improving the Denny^®^ mobile phone application. 

For those aged nine to 12 years, the questionnaire contained additional questions concerning the sources for health information and new information received via the Denny^®^ mobile phone application (*meals/snacking/diet/tooth brushing/oral microbes/xylitol*).

### 2.5. Statistics

For statistical analyses, the children were divided into two age groups, those aged nine to 12 years and those aged four to eight years. The responses were described as frequencies and distributions. The distributions of gender, smart phone ownership, mobile application selection, mobile application costs, brushing habits, main source for oral health information, the use of Denny the Tooth^®^, and usefulness between age groups were evaluated using cross-tabulation and Pearson’s chi-square test or Fisher’s exact test. The distributions of credibility of Denny^®^, and the daily and weekly usage of the application were also compared with Fisher’s exact test. The response variable ‘Does application usage motivate you/your child to brush teeth?’ was analyzed with a logistic regression model in which the explanatory variables were age group, gender, application usefulness, considered the application clear, whether the respondent gained new knowledge about tooth brushing, whether the respondent considered the application challenging, and the average amount of weekly application usage. The explanatory variables were dropped out one by one on the grounds of *p*-value; the final model included only gender and application usefulness. The differences between groups were considered significant when *p* < 0.05. Analyses and graphics were carried out using SPSS software (version 24.0, SPSS, Chicago, IL, USA).

### 2.6. Ethical Considerations

Finnish legislation does not require ethical permission for a survey like the present one [18]. All parents of the participants were volunteers and gave their written consent to use all the information gathered in this study. All analyses were done without personal identification.

## 3. Results

The study population (n = 36) was distributed evenly between those born in 2007 or earlier (group 1) and after 2007 (group 2). The youngest child partaking in the study was four years old, and the oldest were 12 years old (median 9, mean 8.1; SD 1.98). All children were reported to brush their teeth in the evening, with 86.1% also brushing teeth in the morning.

All older children (9–12-year-olds) and 44.4% of the younger ones had their own smart phones. Among the younger group, parents generally decided and chose the mobile applications their children used while among the older ones, the applications to be used were mainly chosen either by the child or mother (Table 1). Most of the applications on the children’s phones were free of charge. When asked why, one parent answered “*That is just our principle. Our children spend too much time with their mobile phones as it is*” while another parent said, “*We don’t want to pay for an application when we can get similar ones free of charge*”. However, almost two thirds of the parents were willing to pay a small fee for an application providing useful and educational activity and half of them considered Denny the Tooth^®^ to be such an application.

Among nine to 12 year-olds, the main source for oral health information were parents and relatives (94.4%). Second, 88.9% of the participants had obtained oral health related information from the dental office. Half of the respondents had learned about oral health at school. One third of the respondents, 33.3%, had received oral health information from TV and radio. Internet and mobile applications (5.6%) had not been important sources for oral health information, nor had community health nurses (22.2%), magazines (5.6%), or friends (5.6%).

In response to the question on new information, 69% of the respondents responded positively. Seventy-two percent reported that they had learned how snacking affects oral health while 42% reported learning about the meaning of proper meals. As many as one third (31%) reported being motivated to reduce snacking after the intervention. Almost 70% answered positively in response to the question “does the application motivate the child or parent in brushing teeth”. Nevertheless, only one third cut back on snacking. The credibility of Denny^®^ was significantly (*p* = 0.016) associated with the application’s daily and weekly usage. 

A majority of the participants considered the application clear (94.4%), amusing (86.1%), and useful (80.6%). Two thirds reported that the application had enough content (Table 2). Three fourths of the respondents found that the Denny the Tooth^®^ application was graphically fascinating. Almost 90% of the respondents used the Denny the Tooth^®^ application several times a week. There were more daily users among the younger ones than the older ones. The overall response to the characteristics of the Denny the Tooth^®^ application were mostly positive. Concerning the Denny the Tooth^®^ character, one parent of a 12-year old child stated “*our 12-year old son didn’t have that much interest in the character because it was too childish, but it still brought more time and thoroughness to daily brushing, which has been difficult to teach otherwise*”.

Approximately 80% of the children used the Denny Timer^®^ while brushing. The same characteristics that described Denny the Tooth^®^ were also reported for the timer (useful and clear). More of the older respondents found the Timer (80%) to be useful than the younger ones (40%) (*p* = 0.040). In more than half of the participants, the application helped to increase the frequency and duration of brushing and in approximately 20% of the participants, made the daily routines easier for the parents. Approximately one third of the parents also saw improvement when the application was not in use and estimated that the improvement could be permanent (Table 3 and Table 4). 

While the participants found the Denny Timer^®^ useful, the odds for brushing were significantly increased (OR = 8.9, 95% CI 1.29–60.60, *p* = 0.026).

## 4. Discussion

This pilot study aimed to investigate the feasibility of a mobile application designed to promote oral health behaviors among preschool and primary school children. The Denny^®^ mobile phone application designed for young children was found to be feasible as the feedback received from the participants was positive and receptive. A majority of the participants found the Denny the Tooth^®^ application to be clear, amusing, and useful. The time spent playing with the application was associated with the credibility of the character. The beneficial outcomes are in line with the few previous studies in this area [19]. The Denny Timer^®^ was reported to improve the quality and time used for brushing as well as the number of those brushing teeth in the morning. As many as one in five reported that evening and morning routines became smoother in the family, even though the tooth brushing habits in the study population were very good. An earlier study also reported that a mobile application can be a motivating tool for routine oral hygiene [9]. Likewise, a recent randomized controlled trail also concluded that a computer game was found to be a tool to deliver oral health education in high-caries-risk children and their families [10].

The strength of this study is the use of a mobile phone application as it keeps reminding, educating, and entertaining the young children. The encouraging results of this study are in line with all previous studies showing the value of text messages and reminders, and that computer games can also be effective [8,10,20]. Additionally, after reviewing the results of this study, it can be presumed that the comprehensive interventional approach as reported by Tsai and colleague can also be effective in this age group [12]. However, this needs further investigation and the application must be continuously updated and developed to be in line with the behavioral theory-based approach. One of the limitations of this study is that the sample size was small. However, the study protocol allowed testing the applications among a small group of children and collecting their perceptions before launching the applications to the application stores. The Denny Timer^®^ was freely available on the Google Play store^®^, but was not developed further, which closed the market. The application must be continuously developed and becomes more challenging and simultaneously rewarding. A larger study population is necessary in future studies. Another limitation is that the study period was also too short for determining changes in oral health behaviors. Finally, the reporting of oral health behaviors, especially among four to eight year olds, was done by the caregivers (proxy informer) in this study. There have also been studies questioning the accuracy of the parents’ perception of their child’s oral health as well as the child’s dental fear [21,22]. Therefore, clinical studies with an objective evaluation of the clinical status before and after the study period would be an important addition to the research frame. 

In this study, more than four fifths of the study participants found that the Denny^®^ mobile phone application was amusing and useful. This shows that there is a demand for easy-to-use health-promoting mobile phone applications of this kind. However, the time spent in front of TVs, computers, and mobile phone applications should be considered during the development. The use of gaming applications has decreased physical activity, particularly among teens and at the same time, there has been a trend toward snacking instead of proper meals [23]. Likewise, a survey also reported that the average screen time among Finnish 7–9-year-old is between two and four hours per day and the average screen time is greater on weekends than schooldays [24]. However, research related to time spent by the customers on the web and mobile phone applications related to health should be studied. 

Although all older children and almost half of the younger ones in this study owned smart phones, only one tenth of them received oral health related information from Internet and mobile phone applications. It should be noted that the web and mobile phone applications offer a great opportunity for oral health promotion [10,25]. A systematic review also reported similar findings on the use of mobile app-based interventions for improving physical activity and diet [26]. Furthermore, the importance of mobile phone applications during the COVID-19 pandemic was observed as most of the health information was disseminated through mobile phone applications [27].

The Denny^®^ character appealed the most to children aged 5–9 years while, one of the participants reported it as being too childish despite improvement in their brushing habits. This shows that the value of the mobile applications is beneficial in daily routines. However, one should bear in mind that the development of game-like applications targeting different age groups is essential. The use of game as an intervention had a significant impact on improving oral health status among Indian schoolchildren [28]. Furthermore, two thirds of the participants in this study were likely to invest in educational/health-related mobile applications like the one used here. This also reflects the motivation of the participants with regard to health and well-being.

A recent survey showed that only 66% of primary school children and 60% of the boys and 73% of the girls among 10–11year-olds in Finland reported brushing their teeth on a twice-a-day basis. Likewise, 26% of Finnish schoolchildren do not eat breakfast every morning on weekdays, and up to 84% of the children skip school lunch [29]. It can be presumed that a mobile phone application can be a useful tool for promoting oral health behaviors, as reported by a recent systematic review [25]. Furthermore, it should also be noted that in this pilot study, morning brushing frequency by young children improved after the use of the mobile phone application. Similarly, they also reported that the mobile phone application was both informative and motivating in terms of diet and snacking. Nevertheless, in further development of mobile applications aimed at oral health promotion, it should be kept in mind that the theoretical basis for behavior changes is necessary as it is known that atheoretical interventions do not work in practice [14].

## 5. Conclusions

In conclusion, the Denny^®^ mobile application designed for young children was found to be feasible based on the feedback received from the participants. The feedback was positive and receptive. Participants also reported an improvement in the quality and time used for brushing. In future, mobile phone applications will have the possibility of promoting oral health in different social and peer groups such as school classes or day care groups as they offer a platform for common goals and comparing results. 

## Figures and Tables

**Figure 1 ijerph-18-03256-f001:**
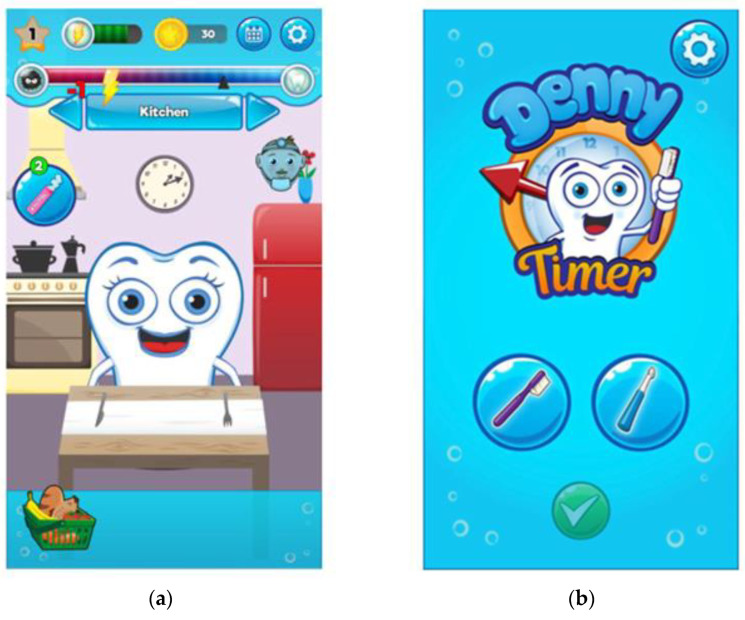
Graphical image of Denny the Tooth^®^ application (**a**) and Denny Timer^®^ application (**b**).

**Table 1 ijerph-18-03256-t001:** Information about the participants and characteristics of mobile application usage grouped according to the age of participants.

	Group 1 (9–12 Year-Olds)	Group 2 (4–8 Year-Olds)	*p* Value
	n (%)	n (%)	
Children	18 (100.0)	18 (100.0)	
Girls	15 (83.3)	11 (61.1)	
Boys	3 (16.7)	7 (38.9)	
The child owns a smart phone	18 (100)	8 (44.4)	<0.001 ^2^
Who selects the mobile applications the child uses?			
Mother	12 (66.7)	18 (100.0)	0.019 ^1^
Father	8 (44.4)	17 (94.4)	0.001 ^1^
Child	13 (72.2)	6 (33.3)	0.019 ^1^
Mobile applications are mainly			
Paid	0	0	
Free	18 (100.0)	18 (100.0)	
Is your family willing to pay a small non-recurring fee for a useful application which offers child educational pastime/entertainment of a good/high quality?	12 (66.7)	15 (83.3)	N.s.
Is Denny the Tooth^®^ such an application?	9 (50.0)	11 (61.1)	N.s.
Have you bought educational games for your child?	6 (33.3)	11 (61.1)	N.s.

^1^ Fischer exact test; ^2^ Chi-Square test; N.s. = non-significant.

**Table 2 ijerph-18-03256-t002:** Respondents’ opinions about Denny the Tooth^®^ application.

How Did You Find Denny the Tooth^®^ Application?	Yes n (%)	No n (%)	No Opinion n (%)	Total n (%)
Clear	34 (94.4)	1 (2.8)	1 (2.8)	36 (100)
Amusing	31 (86.1)	4 (11.1)	1 (2.8)	36 (100)
Useful	29 (80.6)	1 (2.8)	6 (16.7)	36 (100)
Right size/Enough content	24 (66.7)	7 (19.4)	5 (13.9)	36 (100)
Useless	1 (2.8)	32 (88.9)	3 (8.3)	36 (100)
Boring	1 (2.8)	31 (86.1)	4 (11.1)	36 (100)
Fascinating	27 (75.0)	3 (8.3)	6 (16.7)	36 (100)

**Table 3 ijerph-18-03256-t003:** The impact of the Denny Timer^®^ on children’s tooth brushing.

Denny Timer ^®^ Application	Proportion (%) of Those Agreeing with the Statements	
	Yes n (%)	Total n (%)
Taught child to brush teeth better	25 (69.4)	36 (100)
Increased the duration of brushing	25 (69.4)	36 (100)
Made brushing more fun	22 (61.1)	36 (100)
Inspired child to brush teeth for a full two minutes as recommend by dentists	20 (55.6)	36 (100)
Increased child’s initiative to brush teeth	18 (50.0)	36 (100)
Improved child’s tooth brushing routines	15 (41.7)	36 (100)
Made family’s evening routines go smoothly	9 (25.0)	36 (100)
Made family’s morning routines go smoothly	7 (19.4)	36 (100)
It is possible that the Denny Timer^®^ made permanent improvements in brushing	13 (36.1)	36 (100)
Child learned to brush better and longer also when the Denny Timer^®^ was not in use	12 (33.3)	36 (100)

**Table 4 ijerph-18-03256-t004:** Distribution of respondents’ opinions on an instructive oral health mobile application.

Respondents’ Opinions	Yes n (%)	No n (%)	No Opinion n (%)	Total n (%)
Application increased child’s interest in teeth	26 (72.2)	5 (13.9)	5 (13.9)	36 (100)
Application increased duration of tooth brushing	26 (72.2)	9 (25.0)	1 (2.8)	36 (100)
Application helped child to take better care of teeth	22 (61.1)	10 (27.8)	4 (11.1)	36 (100)
Application increased child’s self-motivation to brush his or her teeth	20 (55.6)	12 (33.3)	4 (11.1)	36 (100)
Application increased brushing frequency in the morning	6 (16.7)	28 (77.8)	2 (5.6)	36 (100)
Application increased brushing frequency in the evening	3 (8.3)	32 (88.9)	1 (2.8)	36 (100)
Application taught the meaning of xylitol	20 (55.6)	13 (36.1)	3 (8.3)	36 (100)
Application increased usage of xylitol after meals	7 (19.4)	27 (75.0)	2 (5.6)	36 (100)
Application helped the family with morning and evening routines	17 (47.2)	17 (47.2)	2 (5.6)	36 (100)

## Data Availability

Data are available from the authors upon reasonable request.

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
