# Peer review of "Acceptability of a Mobile Application in Children’s Oral Health Promotion—A Pilot Study"

_ijerph, 2021, doi:10.3390/ijerph18063256_

Round 1
Reviewer 1 Report
This research is under the scope of this journal; the topic is relevant for readers, and this research deals with potentially significant knowledge to the field.
Abstract: please without subheadings
Introduction: I think that the caries is a big problem in special patients too. please consider this reference: PMID: 33419107
What is the importance of this study? You do not think this study is included to the others already done? Which results are comparable?
Please add the null hypothesis in the aim section.
Lack of significance (p <???) in some explanations, in the results sectio
There are many mistakes in the references section and in the text
The discussion is also misleading. What is the novelty of this paper???
Limitations?
Conclusions were not totally supported by the data showed.
Figure legends: Bad descriptions
Author Response
Reviewer 1
This research is under the scope of this journal; the topic is relevant for readers, and this research deals with potentially significant knowledge to the field.
Thank you for your comments. We have tried to address them.
Abstract: please without subheadings
We have now removed the subheadings of the abstract.
Introduction: I think that the caries is a big problem in special patients too. please consider this reference: PMID: 33419107
We have now added this reference and the text as follows:
The restorative treatment need is also high among the children with special health care need [3] in the page 1, line 35-36.
What is the importance of this study? You do not think this study is included to the others already done? Which results are comparable?
Thank you for pointing this out. We have now added the text to the discussion section in page 10 line 287-290 as follows:
An earlier study also reported that a mobile application can be a motivating tool for routine oral hygiene [9]. Likewise, a recent randomized controlled trail also concluded that a computer game was found to be a tool to deliver oral health education in high-caries-risk children and their families [10].
Please add the null hypothesis in the aim section.
We have added null hypothesis as follow in line 82-85:
The null hypothesis for this study is that a mobile phone application designed for preschool as well as primary school children is not acceptable for use and it cannot improve oral health behaviours in short test period.
Lack of significance (p <???) in some explanations, in the results section
Most of the p-values are presented in the table, therefore we did not consider adding them to the text.
There are many mistakes in the references section and in the text
We have carefully revised the text and reference section.
The discussion is also misleading. What is the novelty of this paper???
We have discussed the study findings and methodology as well as compare them with previous studies. For the improvement, we have also considered the reviewers’ comment in the current manuscript.
The aim of this pilot study was to investigate the acceptability of an oral health-related mobile application developed for young children based on the feedback given by the children and their parents. Another aim was to evaluate the self-reported change in children’s oral health behaviours during a short test period of five-weeks.
Furthermore, the target of this study was to further develop the mobile application after the pilot test period based on the feedback and comments received from the study participants.
Limitations?
We have had added a strength and limitation section in our previous version in page 10 line 292-312. The text was as follows:
The strength of this study is the use of a mobile phone application as it keeps reminding, educating, and entertaining the young children. The encouraging results of this study are in line with all previous studies showing the value of text messages and reminders and that computer games can also be effective [8, 10, 20]. Additionally, after reviewing the results of this study, it can be presumed that the comprehensive interventional approach as reported by Tsai and colleague can be also be effective in this age group [12]. However, this needs further investigation and the application must be continuously updated and developed to be in line with the behavioural theory-based approach. One of the limitations of this study is that the sample size is small. However, the study protocol allowed testing the applications among a small group of children and collecting their perceptions before launching the applications to the application stores. Denny Timer® was freely available on Google Play store, but was not developed further, which closed the market. The application must be continuously developed and become more challenging and simultaneously rewarding. A larger study population is necessary in future studies. Another limitation is that the study period was also short for determining changes in oral health behaviours. Finally, the reporting of oral health behaviours, especially among 4- to 8-year olds, was done by the caregivers (proxy informer) in this study. There are studies questioning the accuracy of parents’ perception of child’s oral health as well as the child’s dental fear [21-22]. Therefore, clinical studies with an objective evaluation of the clinical status before and after the study period would be an important addition to the research frame.
Conclusions were not totally supported by the data showed.
We have now revised our conclusion section that is in line with the study objectives and results.
In conclusion, the Denny® mobile application designed for young children was found to be feasible based on the feedback received from the participants. The feedback was positive and receptive. Participants also reported an improvement in the quality and time used for brushing. In future, mobile phone applications will have possibilities to promote oral health in different social and peer groups such as school classes or day care groups as they offer a platform for common goals and comparing results.
Figure legends: Bad descriptions
Graphical image of Denny the Tooth® application (a) and Denny Timer® application (b)

Reviewer 2 Report
In this exploratory study, the aim was:
1) to investigate the feasibility of an oral health-related mobile application developed for young children
2) to evaluate the feedback given by children and their parents and whether the application had an impact on the children’s oral health behaviours during a short test period.
Both aims are answered in this work, however my major concern is who has been surveyed (I believe it was the parents, but it is not clear when were the parents or the children). Please, clarify this point throughout the manuscript.
Furthermore, editing of English language and style are required.
Concerning the work itself is interesting and with potential, though it would be more interesting to both present the exploratory and confirmatory data.
Author Response
Reviewer 2
In this exploratory study, the aim was:
1) to investigate the feasibility of an oral health-related mobile application developed for young children
2) to evaluate the feedback given by children and their parents and whether the application had an impact on the children’s oral health behaviours during a short test period.
Both aims are answered in this work, however my major concern is who has been surveyed (I believe it was the parents, but it is not clear when were the parents or the children). Please, clarify this point throughout the manuscript.
Thank you for your encouraging comments. We have added this information in our current manuscript in page 4 line 199.
After five weeks, all phones were returned, and while doing so, the participants (i.e. caregiver) received a link to a Webropol-based survey.
Furthermore, editing of English language and style are required.
We have completed the professional English proof reading in our current version.
Concerning the work itself is interesting and with potential, though it would be more interesting to both present the exploratory and confirmatory data.
Thank you for raising this question, we tried to perform the principal component analysis to analyse the construct of the questionnaire. However, the KMO was < 0.6 in most of the factors, therefore, giving an ill-conditioned eigenvalue. which can be due to too small sample size for this kind of analysis.

Reviewer 3 Report
This manuscript reports findings from a small study on the feasibility of using a mobile application with children. Some comments to address:
The study doesn’t have ethics approval. This is worrisome because you are giving mobile phones to children as young as 4 years old (vulnerable group). Not sure about the legislation in the country this study was conducted, but in the EU ethics approval is required for any study in humans. This needs a closer look by the editorial office.
Revise the aim of the study, you are not measuring an “improvement in dental behaviours” as stated in introduction (presumably that would be done in a definite trial at a later stage) so please be concrete about what you expected to find in this small study.
It would be helpful to describe the full project (feasibility->pilot->definite trial) at the beginning of Methods to give readers an idea of what is the contribution of this small study to the overall design.
Another important missing point is the theoretical underpinning for the development of the application. What behaviour change theory was used to develop the app? There is now strong evidence that atheoretical interventions (either simple or complex) will not work in practice. Please present this information, if available, in the introduction.
The words feasibility and pilot are used very lightly throughout the manuscript. Pilot and feasibility studies are not the same (see PMID: 24735841 and 20637084), but you used the terms interchangeably. What was your definition of feasibility? And more importantly what are the pre-set criteria used to decide if the intervention was feasible or not? This is important to decide whether you can progress to the next stage of the full project (see previous comment). I have got the impression, from the way the results are presented, that you measured acceptability (feedback?) not feasibility.
There are international guidelines to report findings from feasibility and pilot studies (not necessarily trials) and you to adhere to them (PMID: 27777223).
Author Response
Reviewer 3
This manuscript reports findings from a small study on the feasibility of using a mobile application with children. Some comments to address:
The study doesn’t have ethics approval. This is worrisome because you are giving mobile phones to children as young as 4 years old (vulnerable group). Not sure about the legislation in the country this study was conducted, but in the EU ethics approval is required for any study in humans. This needs a closer look by the editorial office.
According to the Finnish legislation (Medical research act no 488/1999), cross-over surveys (without any personal identification and/or clinical record) do not require ethical permission.
Furthermore, we aimed to receive feedback concerning the Denny® mobile phone-application that primarily involves survey questions mentioned in the study. The smart phones used in this study were provided by the research team without SIM cards and could not be used for making phone calls. The phone model was Alcatel Pixi 4® with both Denny® applications installed by the research team. Furthermore, children could use the mobile phone under the supervision of parents. In this study, parents also gave their written consent for their participation.
Revise the aim of the study, you are not measuring an “improvement in dental behaviours” as stated in introduction (presumably that would be done in a definite trial at a later stage) so please be concrete about what you expected to find in this small study.
We have revised our study aims as follows:
The aim of this pilot study was to investigate the acceptability of an oral health-related mobile application developed for young children based on the feedback given by the children and their parents. Another aim was to evaluate the self-reported change in children’s oral health behaviours during a short test period of five-weeks.
It would be helpful to describe the full project (feasibility->pilot->definite trial) at the beginning of Methods to give readers an idea of what is the contribution of this small study to the overall design.
We have now changed the place of the 2.1. study design at the beginning of the method section followed by the 2.2. study participants.
Another important missing point is the theoretical underpinning for the development of the application. What behaviour change theory was used to develop the app? There is now strong evidence that atheoretical interventions (either simple or complex) will not work in practice. Please present this information, if available, in the introduction.
Thank you for pointing this out.
We have now added a paragraph in the introduction section page 2 line 69-77 as follows:
Behavioural change is an important aspect of oral health promotion [14]. Self-efficacy is considered to be an individual’s belief in his or her capacity to execute behaviours necessary to provide specific performance for attainment. In this model, mastery experiences, vicarious experiences, verbal persuasion, and emotional arousal can be the sources of information to be motivated for goal attainment [15]. There is evidence concerning the influence of theory-based behavioural change on dental caries reduction [16]. The parental role is crucial in adopting healthy habits including oral hygiene. A systematic review reported that maternal self-efficacy was associated with dental caries in children [17].
We have also added a paragraph in the discussion section page 2 line 69-77 as follows:
Nevertheless, in further development of mobile applications aimed at oral health promotion, it should be kept in mind that the theoretical basis for behaviour changes is necessary as it is known that atheoretical interventions do not work in practice [14].
The words feasibility and pilot are used very lightly throughout the manuscript. Pilot and feasibility studies are not the same (see PMID: 24735841 and 20637084), but you used the terms interchangeably. What was your definition of feasibility? And more importantly what are the pre-set criteria used to decide if the intervention was feasible or not? This is important to decide whether you can progress to the next stage of the full project (see previous comment). I have got the impression, from the way the results are presented, that you measured acceptability (feedback?) not feasibility.
Thank you for clearing out this topic. We have now used the term ‘pilot study’ through out the manuscript to avoid confusion to the readers. Indeed, we measured acceptability, and, our aim was to further develop the mobile application after this pilot study. After taking account of this, we have now changed our manuscript title as “Acceptability of a mobile application in children’s oral health promotion – a pilot study”
There are international guidelines to report findings from feasibility and pilot studies (not necessarily trials) and you to adhere to them (PMID: 27777223).
Thank you for the recommendation. We have revised our manuscript as suggested.

Round 2
Reviewer 2 Report
Thank you for your revision. All remarks were resolved properly.
Reviewer 3 Report
I am satisfied with the revisions made by the authors.